# Transferable Contextual Network for Rural Road Extraction from UAV-Based Remote Sensing Images

**DOI:** 10.3390/s25051394

**Published:** 2025-02-25

**Authors:** Jian Wang, Renlong Wang, Yahui Liu, Fei Zhang, Ting Cheng

**Affiliations:** 1Faculty of Information Engineering and Automation, Kunming University of Science and Technology, Kunming 650504, China; 20222204274@stu.kust.edu.cn (R.W.); 20222104064@stu.kust.edu.cn (Y.L.); 20222104055@stu.kust.edu.cn (F.Z.); 20222204326@stu.kust.edu.cn (T.C.); 2Yunnan Key Lab of Artificial Intelligence, Kunming University of Science and Technology, Kunming 650504, China

**Keywords:** remote sensing, rural road extraction, semantic segmentation, Stable Diffusion

## Abstract

Road extraction from UAV-based remote sensing images in rural areas presents significant challenges due to the diverse and complex characteristics of rural roads. Additionally, acquiring UAV remote sensing data for rural areas is challenging due to the high cost of equipment, the lack of clear road boundaries requiring extensive manual annotation, and limited regional policy support for UAV operations. To address these challenges, we propose a transferable contextual network (TCNet), designed to enhance the transferability and accuracy of rural road extraction. We employ a Stable Diffusion model for data augmentation, generating diverse training samples and providing a new method for acquiring remote sensing images. TCNet integrates the clustered contextual Transformer (CCT) module, clustered cross-attention (CCA) module, and CBAM attention mechanism to ensure efficient model transferability across different geographical and climatic conditions. Moreover, we design a new loss function, the Dice-BCE-Lovasz loss (DBL loss), to accelerate convergence and improve segmentation performance in handling imbalanced data. Experimental results demonstrate that TCNet, with only 23.67 M parameters, performs excellently on the DeepGlobe and road datasets and shows outstanding transferability in zero-shot testing on rural remote sensing data. TCNet performs well on segmentation tasks without any fine-tuning for regions such as Burgundy, France, and Yunnan, China.

## 1. Introduction

Remote sensing imagery captured by UAVs is a treasure trove of spatial structure, geographical data, and geometric texture features. Road information, as a pivotal element within these images, is extensively leveraged in urban planning, military operations, disaster relief, and numerous other domains. In rural areas, characterized by intricate terrains and sparse populations, traditional ground survey methods frequently fall short in acquiring comprehensive information. Consequently, the extraction and analysis of road information from UAV-based remote sensing images become particularly critical, enhancing data acquisition efficiency and accuracy, and providing robust support for various applications, especially in emergency response and resource allocation.

Despite its importance, road extraction from remote sensing images in rural settings remains relatively underexplored. Early traditional methods [1,2] often incurred high computational costs or were exceedingly sensitive to environmental changes. Over the past decade, the advent of convolutional neural networks has led to a series of models for road extraction [3,4,5]. However, these models still encounter significant challenges in practical applications, notably ineffective feature extraction and limited transferability. To address these challenges, we have designed a network specifically for rural road extraction from remote sensing images. This network not only achieves superior road segmentation results but also exhibits robust transferability to other rural remote sensing datasets when trained on public datasets. Our research conducts an analysis from the following perspectives:i.In processing remote sensing image data, the performance of neural networks often degrades due to insufficient or poor-quality samples, known as the few-shot problem. Particularly in rural road studies, most research relies on datasets such as DeepGlobe and Massachusetts, which already suffer from data bottlenecks [6]. While these datasets perform well on open-source platforms, their road extraction capabilities diminish significantly in complex environments, and road interference has not been adequately addressed [7]. Expanding the data is essential, but acquiring remote sensing images is prohibitively expensive, requiring advanced UAV equipment, local government support, and manual road annotation.ii.Over the past decade, convolutional neural networks have dominated semantic segmentation in remote sensing images. However, standard convolution layers have notable disadvantages. Rural roads often exhibit characteristics such as curves, narrow paths, and vegetation overgrowth, making it challenging for standard convolution layers to capture long-distance spatial relationships. Recently, the success of Transformers [8] has brought attention mechanisms to the forefront. Numerous studies have demonstrated that attention mechanisms are highly effective for road extraction in remote sensing images [9,10,11]. They integrate widely distributed feature information, effectively handling long-distance relationships between pixels, such as those between continuous road segments. For rural roads, scattered and set in complex environments, attention mechanisms can more accurately identify roads by incorporating contextual clues.iii.Geographical and climatic differences result in unique topographical features in rural areas across different countries. For instance, terraced fields are prevalent in rural China, whereas rural areas in Europe and the United States may feature flat farmlands and vast pastures. These topographical features pose significant challenges for road extraction from remote sensing images. The geographical differences hinder models trained on one dataset from performing well when transferred to others, primarily due to the insufficient optimization of their generalization capabilities. Existing research often focuses on improving performance for specific tasks, with relatively few studies addressing the generalization capability and applicability of models.

To achieve a high-transferability and high-accuracy application for rural road extraction, we propose a transferable contextual network (TCNet), specifically designed for road extraction in rural areas from remote sensing images. The main contributions of this work are structured as follows:We utilized the Stable-Diffusion-2-Inpainting model [12] to generate new rural remote sensing images based on the labels of the public DeepGlobe dataset for data augmentation. This method effectively increases the diversity of training samples.We extended shallow features using the clustered contextual Transformer (CCT) for capturing long-distance dependencies and the clustered cross-attention (CCA) module for optimizing global contextual information and computational efficiency in rural road extraction.We designed a dynamic hybrid loss function, Dice-BCE-Lovasz loss (DBL loss), to improve the accuracy of segmentation in unbalanced samples.

TCNet demonstrated significant performance improvements on two datasets and exhibited robust segmentation capabilities on other UAV-based remote sensing data in a zero-shot context. To emphasize the exceptional transferability of TCNet, Figure 1 shows an aerial view from the movie *The Hobbit: An Unexpected Journey*. Even though this image is not part of any training dataset, TCNet successfully extracts roads without fine-tuning. This highlights the robustness of our model in unseen environments.

## 2. Related Work

### 2.1. Approaches for Semantic Segmentation

Road extraction from remote sensing images fundamentally involves a semantic segmentation task, classifying each pixel as either road (1) or non-road (0). In recent years, the rapid advancements in deep learning technologies have significantly enhanced the efficiency and accuracy of remote sensing image analysis tasks [13,14,15,16,17]. Deep learning-based road segmentation methods can be broadly categorized into three types: fully convolutional network (FCN)-based structures, self-supervised structures based on generative models, and the prevalent encoder-decoder architectures.

FCN-based methods have demonstrated robust performance in road extraction tasks [18,19,20]. However, the asymmetry between the encoder and decoder in FCNs leads to a substantial disparity in the number of parameters between these components. This asymmetry, coupled with the neglect of high-resolution feature maps, often results in the loss of edge information. Consequently, FCN networks struggle to adapt to new data distributions, especially those featuring irregular roads and complex terrains typical of rural areas.

Self-supervised structures based on generative models, notably generative adversarial networks (GANs), can effectively compensate for the lack of road feature information. The adversarial structure of GANs facilitates stable network convergence [21,22]. Several GAN-based methods have performed admirably on public road datasets [22,23,24,25,26]. For example, Zhang, Han, Li, Tang, Zhou, and Jiao [25] proposed an improved GAN for road extraction that requires only a small number of samples for training and achieved high pixel accuracy on the Massachusetts Road Dataset. Zhang, Xiong, Zang, Wang, Li, and Li [26] introduced a multi-scale GAN-based road extraction method that jointly trains spectral and topological features. Nevertheless, these GAN-based methods still encounter challenges such as gradient vanishing and mode collapse, leading to information loss. Moreover, the generator might rely excessively on specific features of the training data to produce accurate outputs, significantly limiting the model’s transferability.

In recent years, diffusion models have emerged as powerful generative models, showing great potential in image generation, inpainting, and segmentation tasks. MaskDiffusion [27] proposed a method that leverages pre-trained diffusion models for semantic segmentation, significantly improving segmentation accuracy, particularly in complex backgrounds or situations with incomplete annotations. Diffusion models generate high-quality segmentation masks through a reverse process, effectively enhancing the performance of traditional models on certain remote sensing datasets. Additionally, Mosaicfusion [28] utilizes diffusion models as an augmentation tool, generating diverse data samples required for large-scale instance segmentation, which successfully enhances the performance of instance segmentation models in scenarios with large vocabularies. Although diffusion models can significantly improve segmentation performance, they require substantial computational resources during the generation process. Moreover, in cases of scarce annotations, the generated masks may contain errors, leading to unstable segmentation results.

Encoder-decoder models represent the most common structure in road extraction tasks [9,29,30,31,32,33]. However, as network depth increases, the input information may become diluted, leading to the loss of road details and reduced extraction accuracy. To address this issue, some researchers have attempted to enhance feature fusion. For instance, Zhou, Zhang and Wu [31] employed atrous convolution to connect the encoder and decoder, achieving first place in the CVPR DeepGlobe 2018 Road Extraction Challenge. However, the sparse connections inherent in atrous convolution may result in certain detailed features not being adequately captured, still causing information loss.

Multi-scale segmentation designs enhance the adaptability of segmentation models to various objects in remote sensing images by incorporating features at different scales, especially for objects like roads that may appear at varying scales. Many methods have adopted multi-scale strategies to improve segmentation performance. For instance, the Boundary-aware feature propagation method [34] introduces a boundary-aware mechanism that enhances feature propagation in boundary regions, allowing for precise segmentation of complex object boundaries. However, although multi-scale designs effectively capture features at different scales, they may face issues of information loss when processing smaller scales or complex backgrounds.

Building on these methods, the introduction of attention mechanisms has significantly improved encoder-decoder structures. For instance, Ren, Yu, and Guan [9] designed multi-scale contextual enhancement and two types of feature attention modules to increase the accuracy of remote sensing road segmentation. Li, Qiu, Chen, Mei, Hong, and Tao [33] integrated lightweight spatial and channel attention modules that can adaptively refine features to handle the complexity of geographical elements. While attention mechanisms have undoubtedly improved road extraction accuracy and enhanced model transferability, they still present challenges. Calculating the self-attention matrices for query, key, and value incurs a substantial computational burden, particularly when training on high-resolution images or large datasets. Thus, although attention mechanisms excel in improving feature representation and segmentation accuracy, their computational complexity and resource requirements remain critical challenges that need to be addressed.

### 2.2. Approaches for Data Augmentation

In the task of road extraction from remote sensing images, data augmentation methods play a crucial role in improving the generalization ability and robustness of models. Traditional data augmentation methods involve single-sample transformations, such as geometric transformation [35], sharpness transformation [36], noise disturbance [37], and random erase [38]. We believe that traditional data augmentation methods generate training samples that significantly deviate from real-world scenarios, thereby limiting the portability of the models. Enhancement techniques that closely align with natural conditions are vital for improving model performance across different datasets.

In recent years, with the enhancement of computational power, deep generative models have made significant progress in the field of data augmentation. The variational auto-encoder (VAE) can effectively expand datasets by generating diverse and robust images, retaining more detailed information during the generation process. For instance, Elbattah et al. [39] utilized VAE to generate high-quality traffic accident data and medical imaging data. However, images generated by VAE are often blurry because the balance between reconstruction loss and KL divergence leads to the loss of detail in the generated images, which limits the wide application of VAE in remote sensing image data augmentation.

Generative adversarial networks (GANs) are the most frequently used image generation technique in the field of computer vision, particularly in remote sensing image generation [40,41,42], due to their simple model architecture and excellent results. Liu, Yang, Deng, Qian, and Fan [41] processed images of different resolutions through the U-NET structure and combined the CBAM module to adaptively adjust spatial and channel features, generating high spatio-temporal resolution remote sensing images. Rui, Cao, Yuan, Kang, and Song [42] generated multiple remote sensing images using a disaster translation GAN, with the damaged building GAN generating specific affected areas through mask-guided generation, used for generating remote sensing images of multiple disaster types. These studies demonstrate the potential of GANs in remote sensing image generation and data augmentation. However, the training process of GANs is prone to gradient vanishing and mode collapse issues. Additionally, GANs heavily rely on specific features of the training data and are highly sensitive to dataset quality. Most importantly, well-trained GAN models can typically only generate images similar to the training set, lacking the ability to generate more diverse images.

To overcome these issues, diffusion models have been introduced into data augmentation for remote sensing images [43]. Diffusion models generate more diverse and robust images by gradually adding noise and then generating images in reverse. Sebaq and ElHelw [43] generated low-resolution images from text using a low-resolution generative diffusion model (LR-GDM) and then enhanced them to high resolution using a super-resolution diffusion model (SRDM). However, there remains an issue: the generated images still require manual annotation for further work, especially semantic segmentation. Although we have methods to generate high-quality images, the subsequent annotation work still entails high costs. Therefore, if we can combine text features and the original image’s label features, altering only the background information of the images, it would be a highly efficient approach for practical research applications.

## 3. Methodology

To construct a highly accurate and transferable network for rural road extraction, we propose TCNet (transferable contextual network), which adopts an encoder-decoder structure in its overall architecture. The overall architecture of TCNet is depicted in Figure 2.

To enhance the model’s robustness, we use a Stable Diffusion model to generate remote sensing images with backgrounds resembling rural areas, based on designed prompts and labels from the DeepGlobe dataset. These synthetic images are combined with data from public datasets to mitigate the issue of insufficient training data.

In the encoder phase, we integrate four clustered contextual Transformer (CCT) modules and introduce two clustered cross-attention (CCA) modules at the encoder-decoder junction for feature fusion. Additionally, we employ the convolutional block attention module (CBAM) in the skip connections between the encoder and decoder to ensure that crucial features from the encoder are preserved and seamlessly integrated into the decoder. This approach enhances feature extraction and segmentation performance in the complex and diverse environments typical of rural areas.

### 3.1. Clustered Cross-Attention

We propose the clustered cross-attention (CCA) module, characterized by its integration of grouped convolution and layer normalization to optimize global contextual information capture and computational efficiency. The design of the CCA module is inspired by CCNet [44]. The primary purpose of the CCA module is to address the challenges in rural road extraction tasks, particularly in complex and variable backgrounds of rural remote sensing images.

As shown in Figure 3, the clustered cross-attention (CCA) module integrates grouped convolution and layer normalization to efficiently capture global contextual information, addressing challenges in rural road extraction from complex backgrounds in remote sensing images. Given an input feature map $H\in R^{H\times W\times C}$, the module first generates the query, key, and value feature maps using grouped 1 × 1 convolutions. The attention-weighted output feature map Y is computed as:(1)$$\;Y=H+\mathrm{softmax}\left(\mathrm{LayerNorm}\;(QK^{⊤})\right)V$$
where it combines the contextualized feature representation with the original input, ensuring that the CCA module effectively captures global context while maintaining computational efficiency, making it highly suitable for rural road segmentation in remote sensing applications.

### 3.2. Clustered Contextual Transformer

Traditional convolutional self-attention mechanisms in Transformers face challenges in rural road extraction due to their independent handling of each query–key pair, which overlooks the broader contextual relationships necessary for interpreting the complex and irregular patterns typical in rural terrains. These mechanisms fail to effectively integrate these contextual pairings, which is crucial for identifying and segmenting the non-uniform and intertwined road features prevalent in rural areas.

As shown in Figure 4, the clustered contextual Transformer (CCT) is developed to overcome the limitations of traditional convolutional self-attention mechanisms in rural road extraction, particularly in scenarios where independent handling of each query–key pair fails to capture broader contextual relationships. Given an input feature map $X\in R^{H\times W\times C}$, where H, W, and C represent the height, width, and number of channels of the feature map, respectively, the CCT block begins by applying a standard convolution operation:(2)$$X_{\mathrm{conv}}=\mathrm{Conv}\left(X,k\right)$$
where k is the convolution kernel size. This operation captures local spatial features, yielding an output of $X_{\mathrm{conv}}\in R^{H\times W\times C}$. The feature map is then divided into four clusters using group convolutions, expressed as:(3)$$X_{\mathrm{group}}^{i}=\mathrm{GroupConv}\left(X_{\mathrm{conv}},\;G\right)$$
with G = 4 being the number of groups, which facilitates localized processing and focuses on relevant spatial features within each cluster. The CCT attention mechanism computes the query Q and key K embeddings using 1 × 1 convolutions, maintaining the dimensions for attention calculation. The attention scores are obtained as:(4)$$A=\mathrm{softmax}\left(\frac{QK^{⊤}}{\sqrt{d_{k}}}\right)\;$$
where $QK^{⊤}$ denotes the dot product of the query and key matrices, and $d_{k}$ is the dimension of the key embeddings. The output is then given by O=AV, representing the weighted sum of the value features. To preserve the original dimensions and enhance feature representation, a final operation yields:(5)$$Y={\mathrm{Conv}}_{1\times 1}\left(O\right)+X$$
where the integration of the input X through a residual connection ensures that both localized and global contextual information are effectively captured, thereby enhancing the model’s transferability and robustness in rural road segmentation tasks.

The key contribution of CCT lies in its ability to apply localized group convolutions, which divide the feature map into distinct clusters, thereby enabling focused learning of region-specific features. By leveraging a context-aware attention mechanism, CCT improves the model’s ability to capture both local and global dependencies, essential for interpreting the complex patterns of rural roads.

### 3.3. CBAM Attention

In our approach, the convolutional block attention module (CBAM) [45] is integrated into the skip connections between each encoder and decoder. The CBAM module enhances feature representation by applying spatial and channel attention mechanisms to the output features of the encoder.

Specifically, CBAM sequentially applies channel attention and spatial attention to the encoder output. The adjusted feature maps are then combined with the corresponding decoder layer via skip connections. This integration ensures efficient information transfer and effective fusion of low-level and high-level features. For rural road extraction, CBAM’s attention mechanisms facilitate the identification of key road features within complex backgrounds, thereby enhancing the model’s ability to capture road details accurately.

### 3.4. Stable-Diffusion-2-Inpainting for Data Augmentation

To enhance the rural road extraction task, we used Stable Diffusion to generate image “variants” by selectively inpainting only the background elements of input images. This approach preserves the road features—critical for segmentation—while modifying the background to increase dataset variability. For example, in a rural scene with a winding dirt road, dense forests, and fields, the road portion remains intact while the background (such as vegetation or terrain) is redrawn. We use prompt words like “high resolution, satellite image” and negative prompts like “road, highway” to ensure that no new roads are generated, allowing the model to focus on altering the background.

We conditioned the Stable Diffusion model using a pre-trained multimodal encoder, OpenCLIP-ViT, which translates text prompts into feature embeddings. These embeddings guide the diffusion process, which refines an initial noisy image to align with the desired content. This allows for the creation of diverse yet realistic backgrounds while maintaining road structures.

This method effectively addresses the data bottleneck problem often encountered in remote sensing datasets. By generating realistic variations of background features, we can augment the dataset with a more diverse range of scenarios without requiring manual annotation. The resulting augmented dataset improves the model’s training performance, especially in areas where annotated road data are scarce or difficult to obtain. Moreover, the inpainting technique offers a more natural alternative to traditional augmentation methods like flipping, rotation, or color adjustments, which can often create unrealistic variations. Stable Diffusion, in contrast, generates contextually relevant background modifications that better reflect real-world conditions in satellite imagery, leading to improved model generalization, particularly in rural areas with diverse terrains and inconsistent road patterns.

The structure of the Stable Diffusion generation process is shown in Figure 5, which illustrates the progression from initial noise generation to final background inpainting, ensuring that the model focuses on generating background elements that complement the existing road features and ultimately enhances segmentation performance across different datasets.

### 3.5. Loss Function

In this study, we employed a combination of three different loss functions to form a comprehensive loss function: binary cross-entropy (BCE) loss, soft Dice loss, and Lovász-softmax loss. Each of these loss functions has its own advantages, and by combining them, we can more comprehensively optimize the model’s performance.

The road extraction task in TCNet can be viewed as a binary classification problem (i.e., “road” and “non-road”); thus, we utilized the binary cross-entropy (BCE) loss function.(6)$$\mathrm{BCE}\;\mathrm{Loss}\;=-\frac{1}{N}\underset{i}{\sum }\left[y_{i}\cdot \log\left(p_{i}\right)+\left(1-y_{i}\right)\cdot \log\left(1-p_{i}\right)\right]$$
where ${\;y}_{I}$ is the sample value, with 1 for a positive class and 0 for a negative class, and $p_{I}$ is the predicted value, ranging between (0, 1).

The soft Dice loss is a similarity measure-based loss function particularly suitable for imbalanced data, such as in our road extraction task, where the road areas might constitute only a small portion of the image. Soft Dice loss calculates the loss by measuring the overlap between the predicted results and the true labels, helping the model better capture the minority class features.(7)$$\mathrm{Dice}\;\mathrm{soft}\;\mathrm{loss}\;=1-\overset{N}{\underset{1}{\sum }}\frac{2y_{i}p_{i}+\varepsilon }{y_{i}+p_{i}+\varepsilon }$$
where $y_{I}$ represents the true label of the i-th pixel, $p_{I}$ represents the predicted probability, N is the total number of pixels, and ε is a smoothing term to avoid division by zero.

Although the above two loss functions already offer acceptable performance, they still have issues such as slow convergence and instability in training small targets. Therefore, to further enhance the model’s performance in handling severely imbalanced classes and complex backgrounds in road extraction tasks, we introduced the Lovász-softmax loss. This loss function, based on the Lovász extension, is specifically designed to directly optimize the intersection over union (IoU) metric, making it particularly suitable for performance improvement in semantic segmentation tasks.(8)$$\mathrm{Lovász-softmax}\;\mathrm{Loss}=\frac{1}{c}\sum_{i=1}^{c}\;\mathrm{Lovász-Hinge}\;\left(m_{i}\right)$$
where $m_{i}$ is the error margin for the i-th class. This loss function converts the IoU error of a single sample into a differentiable cost function, directly optimizing the IoU, which is especially useful for small target detection in imbalanced datasets.

By combining the above three loss functions, we defined a comprehensive loss function to simultaneously optimize these critical performance metrics during training. The comprehensive loss function is as follows:L = BCE loss + Dice soft loss + Lovász-softmax loss(9)

## 4. Experiments and Results

### 4.1. Dataset

In this study, we conducted experiments using two datasets: the DeepGlobe Remote Sensing Road Dataset and the Massachusetts Road Dataset. (1) The DeepGlobe Remote Sensing Road Dataset, part of the CVPR 2018 competition, comprises satellite images from diverse global regions, encompassing complex terrains such as urban, rural, and mountainous areas. Each image measures 1024 × 1024 pixels. Due to the unavailability of validation and test set labels, we partitioned 20% of the original training set to create a test set. The original training set consists of 6226 images, with 80% retained for training and 20% allocated for testing. All images were uniformly converted to the RGB three-channel format and standardized to balance brightness and contrast. (2) The Massachusetts Road Dataset comprises 1171 aerial images from the state of Massachusetts, USA. Each image measures 1500 × 1500 pixels, covering an area of 2.25 km^2^. This dataset is publicly accessible at https://www.cs.toronto.edu/~vmnih/data/ (accessed on 3 December 2024). For this study, we employed the Massachusetts Road Dataset for zero-shot validation.

### 4.2. Network Training

The proposed TCNet model was trained end-to-end using stochastic gradient descent (SGD) with backpropagation on a cloud computing platform equipped with an NVIDIA GeForce RTX 3060 GPU with 12 GB of VRAM (NVIDIA, Santa Clara, CA, USA), a 16-core CPU, and 16 GB of memory. The batch size was set to 2 images per GPU, and the model was trained for a total of 200 epochs. The initial learning rate was set to 0.0002 for the first 150 epochs, which was then decayed to 0.00002 for the remaining epochs. The Adam optimizer with β1 = 0.9 and β2 = 0.999 was used, and the learning rate was adjusted according to the performance on the validation set. To augment the training set, several data augmentation techniques were applied, including random horizontal and vertical flips, random 90-degree rotations, and random shifts, scales, and rotations to simulate different geometric transformations. Additionally, random hue, saturation, and value adjustments were applied to simulate variations in illumination conditions. Each original image was transformed into 24 augmented images, increasing the diversity of the training set and enhancing the model’s robustness. The model was validated after each epoch using a validation set that accounted for 20% of the total dataset. Early stopping was implemented to prevent overfitting, and the best-performing model based on validation loss was saved.

To provide a reference for hardware performance, we also measured the giga floating point operations per second (Gfops) during a simple benchmark run on the same NVIDIA GeForce RTX 3060 GPU. The experiment involved performing 10,000,000 floating point operations and measured an execution time of 0.116609 s, resulting in an estimated Gfops value of 0.085756 Gfops. Although this result does not directly correspond to the specific operations performed during model training, it provides a rough estimate of the computational capability of the hardware used in this study, which can help assess the general efficiency of the system.

### 4.3. Model Evaluation Criteria

In this study, we selected four accuracy metrics to evaluate the effectiveness of road extraction: precision, recall, F1 score, and intersection over union (IoU). The definitions of these metrics are as follows:

Precision measures the proportion of correctly identified road areas in the result image relative to the road areas in the annotated image. It is calculated as:(10)$$\mathrm{Precision}=\frac{\mathrm{TP}}{\mathrm{TP}+\mathrm{FP}}$$
where TP (true positive) represents the area predicted as road and actually being road and FP (false positive) represents the area predicted as road but actually not being road.

Recall measures the proportion of correctly identified road areas in the result image relative to the total road areas in the annotated image. It is calculated as:(11)$$\mathrm{Recall}=\frac{\mathrm{TP}}{\mathrm{TP}+\mathrm{FN}}$$
where FN (false negative) represents the area annotated as road but predicted as non-road.

F1 Score is a statistical measure used to evaluate the accuracy of a binary classification model. It is the harmonic mean of precision and recall:(12)$$F1\;\mathrm{Score}=2\times \frac{\mathrm{Precision}\;\times \mathrm{Recall}\;}{\mathrm{Precision}+\mathrm{Recall}}$$

Intersection over union (IoU) describes the overlap between the predicted results and the actual labels. It is defined as:(13)$$\mathrm{IoU}=\frac{\mathrm{Result}\left(\mathrm{road}\right)\cap \mathrm{Label}\left(\mathrm{road}\right)}{\mathrm{Result}\left(\mathrm{road}\right)\cup \mathrm{Label}\left(\mathrm{road}\right)}$$

These metrics are all suitable for evaluating the performance of the model in road extraction tasks, effectively reflecting the model’s ability to handle complex scenarios.

### 4.4. Results and Discussion

#### 4.4.1. Performance Evaluation

In this section, we compare TCNet with five other image segmentation models on the rural road dataset. Precision, recall, F1 score, and IoU are selected to evaluate the road extraction performance. Table 1 reports the quantitative results of the comparison methods on the DeepGlobe dataset.

To verify the effectiveness of road extraction models in remote sensing images, we trained and validated five popular classification networks on the same dataset, with a training and validation ratio of 8:2. The networks used included U-Net [29], DLinkNet [31], NLLinkNet [46], DeepLabV3+ [47], and RCFSNet [48]. These networks are widely regarded as baseline models for comparative studies. The results are summarized in Table 1.

From Table 1, it is evident that although the U-Net model has fewer parameters, its performance in terms of precision, recall, F1 score, and IoU is relatively poor. DLinkNet, which incorporates atrous convolution in the feature fusion part, improves its ability to capture road details due to its structural enhancements. DeepLabV3+ further enhances segmentation accuracy by adopting atrous convolution and an encoder-decoder structure, particularly excelling in IoU compared with other models. However, its increased number of parameters makes the network more complex and unsuitable for lightweight devices such as UAVs. RCFSNet introduces full-stage feature fusion in the skip connections, resulting in three times more parameters than our model, and although its performance is the best, the increase in effectiveness is not proportional to the increase in parameter count, making it impractical for real-world applications.

In contrast, TCNet performs well across all key performance metrics, notably achieving 66.08% IoU and 79.57% F1 score while maintaining a relatively low parameter count (23.67 M). This demonstrates the high efficiency and practicality of our model. These results validate the superiority of our design, effectively utilizing attention mechanisms and the DBL loss function to enhance the model’s adaptability to the severe class imbalance in binary classification problems. The performance improvements, particularly in parameter efficiency and training efficiency, provide an efficient and practical solution for the road segmentation domain.

#### 4.4.2. Transferability Evaluation

To further validate the superiority and applicability of the TCNet model, we conducted a bold experiment by performing zero-shot comparisons on the Massachusetts Road Dataset and directly extracting roads from remote sensing images of rural areas in Yunnan, China, and the Burgundy region in France, without any fine-tuning.

As shown in Table 2, the TCNet model performs remarkably well on the Massachusetts Road Dataset, achieving an F1 score of 53.14% and an IoU of 65.22%, significantly surpassing other models. This result not only underscores TCNet’s effectiveness in handling complex road network structures but also highlights its adaptability to diverse geographical environments.

Moreover, as depicted in Figure 6, we directly applied the trained model to segment roads in remote sensing images of rural areas in Yunnan, China, the Burgundy region in France, Churchill Manor in the UK, and the Alps mountain range. These diverse datasets further assess the model’s transferability across different geographical regions. From the qualitative analysis, TCNet demonstrates superior accuracy in extracting detailed road networks. Compared with other models like DeepLabV3+ and DLinkNet, TCNet exhibits noticeable advantages in extraction accuracy and continuity. This advantage primarily stems from the advanced feature extraction and enhancement techniques employed in TCNet’s network design, enabling it to better identify and reconstruct road information, even in images with complex spectral information or unclear road structures typical of rural areas.

In more densely road-populated regions such as the Alps (i and j), there is no significant difference between TCNet and other models in terms of performance. However, TCNet achieves this level of accuracy with approximately 16 million fewer parameters, demonstrating its efficiency in terms of model complexity.

These results indicate that the TCNet model not only excels on standard test datasets but also exhibits strong cross-region transferability, making it highly suitable for various remote sensing image segmentation tasks. This demonstration of transferability further confirms TCNet’s broad applicability and potential value in practical applications, especially in the domain of remote sensing images of rural areas where environments are variable and road structures are complex.

### 4.5. Ablation Study

#### 4.5.1. Effectiveness of the Three Modules

We conducted an ablation study to evaluate the effectiveness of the proposed modules in TCNet. Specifically, CCA is integrated into the encoder, CCT is utilized for feature fusion, and CBAM enhances the skip connections within the network. To verify the effectiveness of these three modules, we conducted a series of experiments on the DeepGlobe dataset. The quantitative results are presented in Table 3.

The baseline performance, based on DLinkNet, served as a reference point. By incorporating the proposed modules, the F1 score improved from 78.73% to 79.57%, and the IoU increased from 64.92% to 66.08%. These experimental results clearly demonstrate the effectiveness of our proposed modules for road network extraction.

#### 4.5.2. Loss Function Selection

In this section, we evaluated the performance of several different combinations of loss functions on the rural road dataset. Specifically, we tested individual loss functions such as binary cross-entropy (BCE) loss and Dice loss, as well as composite loss functions including BCE loss + Dice loss and Dice-BCE-Lovasz loss (DBL loss). The experimental results were assessed using F1 score and mean intersection over union (mIoU) and are detailed in Table 4.

The results indicate that among the individual loss functions, BCE loss performed better in terms of extraction accuracy, likely due to its emphasis on global performance. Conversely, Dice loss showed superior performance in IoU, primarily because it focuses more on the overlapping regions between image samples. When comparing composite loss functions, both BCE loss + Dice loss and DBL loss outperformed the individual loss functions in terms of accuracy and IoU. This suggests that combining loss functions can effectively enhance model performance.

Notably, DBL loss exhibited the best performance across all tests, achieving the highest accuracy and IoU. DBL loss leverages a dynamic weighting approach, fully exploiting the advantages of combined loss functions. This allows the model to maintain stability while benefiting from the strengths of Dice loss in binary classification problems, improving the model’s adaptability to severe class imbalance situations. Additionally, we observed that the model using DBL loss converged significantly faster during training, reducing the required epochs from 150 to 120. This further demonstrates the effectiveness of DBL loss, as illustrated in Figure 7.

#### 4.5.3. Data Augmentation Evaluation

To enhance the model’s learning capability and generalization with limited training data, and to address the difficulty of acquiring remote sensing images, this study employed a diffusion model to generate new remote sensing image data for dataset augmentation. By utilizing the inpainting functionality of the Stable Diffusion model, we specifically redrew the “background” parts of the images while keeping the “road” parts unchanged, allowing for the use of existing mask images in the original training set as labels. This approach not only increases the diversity of the training data but also maintains the accuracy of the labels. Some of the generated results are shown in Figure 8.

Table 5 presents a quantitative comparison of the original TCNet model and the TCNet model augmented with one thousand generated remote sensing images on the DeepGlobe dataset. As shown in Table 5, on the DeepGlobe dataset, the TCNet model augmented with generated images (TCNet + 1000 generated images) improved the F1 score from 79.57% to 79.69% and the IoU from 66.08% to 66.53%. This indicates that the additional generated images effectively enhanced the model’s ability to capture remote sensing image features.

Similarly, as observed in Table 6, during zero-shot testing on the Massachusetts Road Dataset, the augmented model also showed improvements in F1 score and IoU, increasing from 53.14% and 65.22% to 54.57% and 66.03%, respectively. This increase demonstrates that the images generated by the diffusion model are similar to real remote sensing images, providing a novel approach to addressing the data bottleneck problem in remote sensing imagery.

## 5. Discussion

Why does our designed network exhibit remarkable transferability with minimal parameters? Compared with traditional convolutional networks, the introduction of advanced attention mechanisms effectively increases the model’s performance without significantly adding to the parameter count. Our designed CCA module captures widely distributed contextual information in images, proving highly effective in handling long-distance relationships between continuous road segments in remote sensing images. The CCT module enhances the model’s ability to capture global features while maintaining lower computational complexity compared with various convolution modules. The CBAM module further improves the model’s feature representation capabilities by focusing on important features in both channel and spatial dimensions. The combination of these attention mechanisms not only boosts the model’s performance but also significantly reduces the number of parameters, enhancing the model’s applicability and transferability in resource-limited environments.

To tackle the issue of data scarcity, we employed the Stable-Diffusion-2-Inpainting model to generate new remote sensing image data. This approach not only increases the diversity of the training data but also improves the model’s generalization ability in complex, real-world environments. Particularly for remote sensing image data in highland and rural areas, the generated images are highly consistent with real environments, alleviating the data bottleneck issue in remote sensing data. Finally, we introduced a combination of three loss functions: BCE, soft Dice, and Lovász-softmax, forming the DBL (Dice-BCE-Lovasz) composite loss function. This combination leverages the advantages of each loss function, significantly improving the model’s training speed and accuracy, especially in handling class imbalance and complex background scenarios in road extraction tasks.

Despite these advantages, several limitations remain. Although the model performs well on datasets such as DeepGlobe and the Massachusetts dataset, its performance may degrade when applied to certain types of remote sensing data, particularly in regions with complex urban landscapes or heavily occluded roads. This limitation is partially due to the model’s reliance on the quality and diversity of training data, which, although augmented using Stable-Diffusion-2-Inpainting, still faces challenges when addressing extremely complex environments where traditional road structures may be obscured by vegetation, buildings, or other obstacles. In addition, although the introduction of the DBL (Dice-BCE-Lovasz) composite loss function significantly improves training speed and accuracy, it may still struggle with extreme class imbalance in certain edge cases, such as roads in low-contrast or heavily occluded regions.

We also found that the proposed TCNet performs poorly in areas with deep shadows. Although we increased the remote sensing image data generated by the diffusion model, most of them did not include disturbances like shadows. Therefore, to further improve the model’s generalization, we believe it is necessary to construct a remote sensing dataset covering different geographical locations and environmental conditions. Additionally, some of the images generated using the Stable-Diffusion-2-Inpainting model exhibit regions in the background that resemble road structures, which may interfere with the model’s segmentation performance. This issue is exemplified in Figure 9, where such artifacts can be observed, potentially impacting the accuracy of road extraction. A similar phenomenon was also observed in the DeepGlobe dataset. These findings highlight the need for further development of remote sensing road extraction datasets. Existing public datasets still have limitations, and future advancements in remote sensing image generation techniques may help mitigate data scarcity and the challenges associated with manual mask annotation.

## 6. Conclusions

We proposed a highly transferable network for rural remote sensing road extraction, called TCNet. This model consists of the CCT module, CCA module, and CBAM module and employs the DBL composite loss function. Additionally, we utilized Stable-Diffusion-2-Inpainting for data augmentation, maintaining fixed labels to preserve road information while filling in the background of rural mountainous remote sensing images generated by the diffusion model. This approach facilitates training, even in the absence of readily available remote sensing images or manual annotations.

With these strategies, the proposed TCNet demonstrates excellent transferability and accuracy. It was evaluated on two large remote sensing image datasets, achieving significantly better segmentation results than current mainstream remote sensing road extraction models, despite having only 23.67 M parameters. Under zero-shot conditions, the segmentation performance on other data, such as those from Yunnan, China, and Burgundy, France, was nearly perfect. Experimental results show that our model is capable of practical applications in remote sensing road extraction, effectively handling various complex rural mountainous environments.

Looking forward, there are several avenues for future research. One important direction is to explore how to further improve the model’s robustness in handling highly complex or occluded urban environments. This could involve the integration of more sophisticated attention mechanisms or multi-scale fusion techniques to better capture fine-grained details in such settings. Additionally, expanding the training data to include more diverse environments, particularly highland regions and dense urban areas, would help address the current limitations related to data scarcity and the challenge of generalizing to new geographical contexts. To this end, we are planning to create our own highland-region road dataset, which will help alleviate the data bottleneck issue in road extraction tasks for remote sensing imagery.

## Figures and Tables

**Figure 1 sensors-25-01394-f001:**
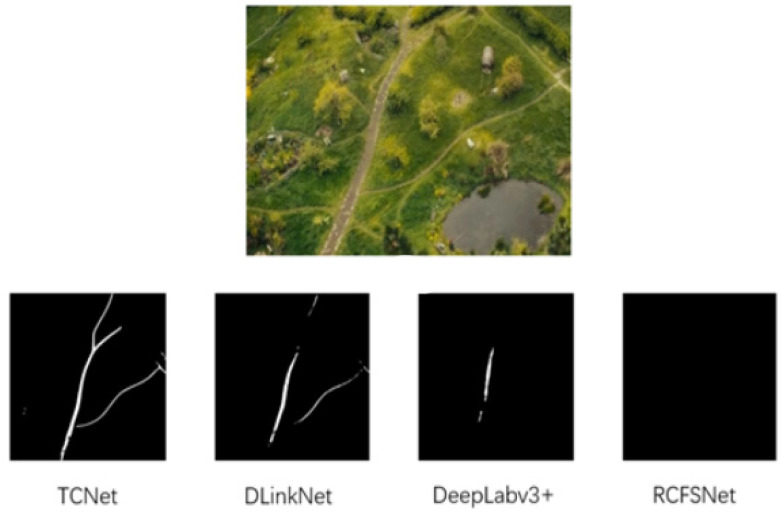
Image from *The Hobbit: An Unexpected Journey* (2012) depicts the Shire as Bilbo follows Gandalf on an adventure. Our proposed TCNet, which was only trained on the DeepGlobe dataset, achieves excellent segmentation of remote sensing images from the movie without any fine-tuning, compared with other models.

**Figure 2 sensors-25-01394-f002:**
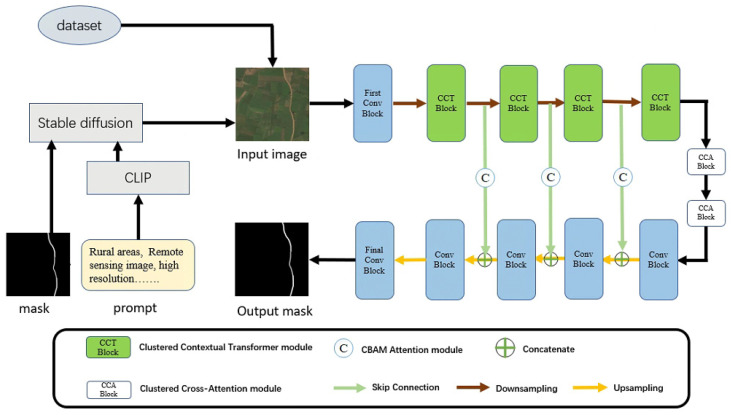
Overall architecture of the proposed TCNet.

**Figure 3 sensors-25-01394-f003:**
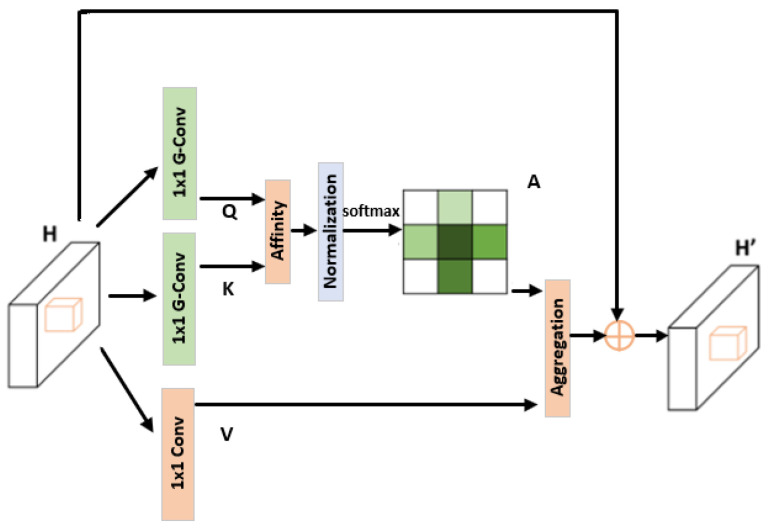
Structure of the clustered cross-attention (CCA) module.

**Figure 4 sensors-25-01394-f004:**
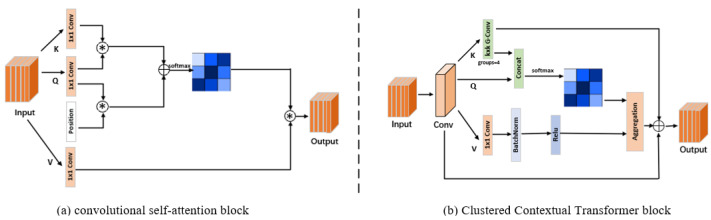
The detailed structures of (**a**) the conventional self-attention block and (**b**) our clustered contextual Transformer (CCT) block. The asterisk (*) represents the dot product operation used to compute attention weights.

**Figure 5 sensors-25-01394-f005:**
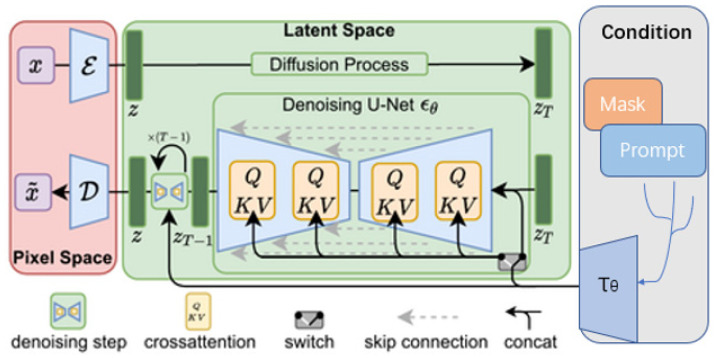
Structure of the stable diffusion generation model.

**Figure 6 sensors-25-01394-f006:**
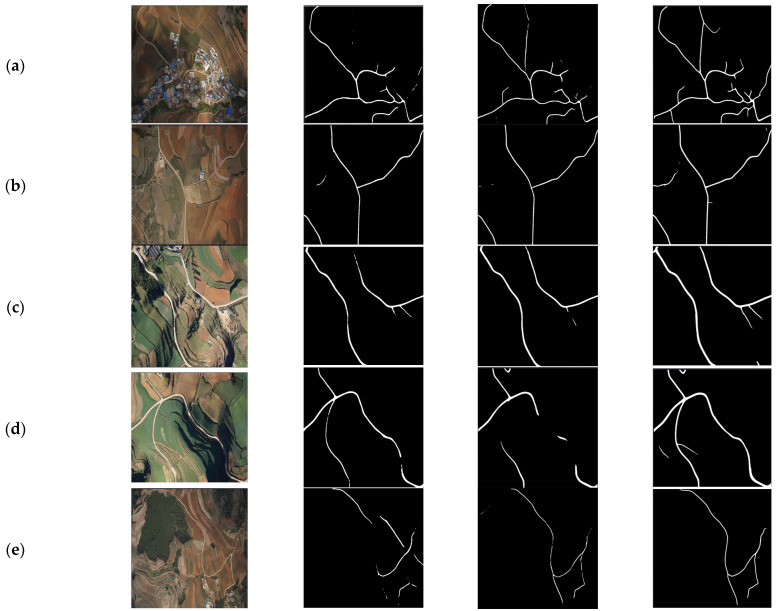
Qualitative results of different road extraction methods on various datasets, demonstrating zero-shot transfer. (**a**–**e**) Remote sensing image of Yunnan Province in China. (**f**,**g**) Remote sensing images of the Burgundy region in France. (**h**) Remote sensing image of Churchill Manor in the UK. (**i**,**j**) Remote sensing images from the Alps mountain range.

**Figure 7 sensors-25-01394-f007:**
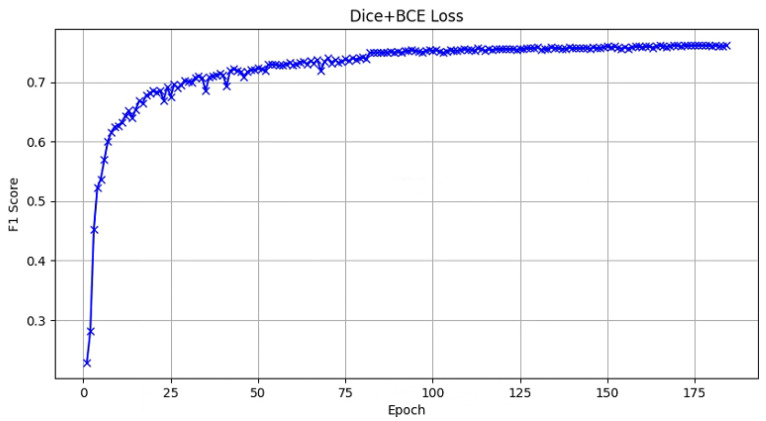
Convergence comparison of loss functions. Compared with the traditional Dice + BCE loss, DBL loss can achieve convergence 60 epochs earlier.

**Figure 8 sensors-25-01394-f008:**
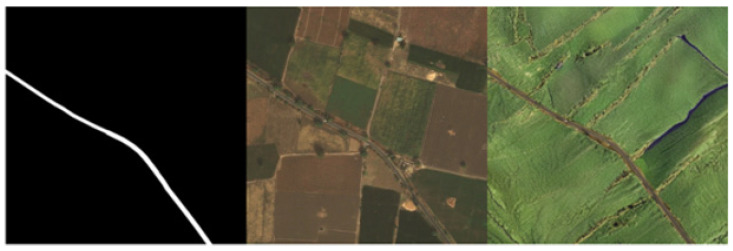
Comparison between the generated image and the original image.

**Figure 9 sensors-25-01394-f009:**
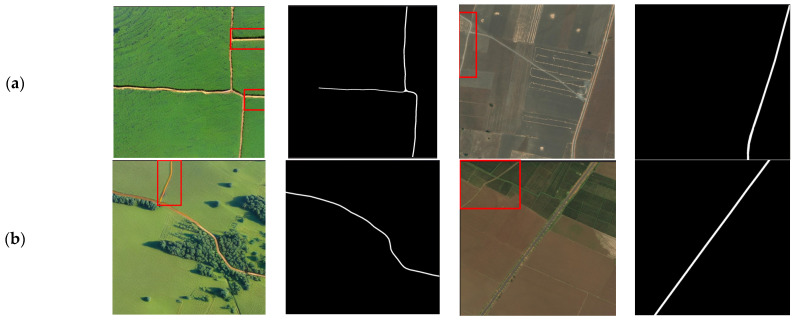
(**a**–**c**) Show the generated remote sensing images alongside the corresponding images from the DeepGlobe dataset. The red box highlights areas where roads were incorrectly generated or where roads are present in the DeepGlobe images but were not labeled. This emphasizes the challenges in ensuring accurate labels in UAV-based remote sensing image datasets.

**Table 1 sensors-25-01394-t001:** Quantitative comparison on the DeepGlobe dataset (%). The bold represents the maximum value per column.

Model	Precision	Recall	F1	mIoU	Parameters (M)
U-Net	75.80	76.90	76.34	61.74	9.85
DLinkNet	77.23	**80.29**	78.73	64.92	31.10
NLLinkNet	79.00	79.46	79.23	65.60	21.82
DeepLabV3+	78.57	79.21	78.89	65.13	39.63
RCFSNet	**81.62**	78.62	**80.09**	**66.79**	**59.28**
TCNet	79.51	79.47	79.57	66.08	23.67

**Table 2 sensors-25-01394-t002:** Zero-shot on the Massachusetts Roads dataset (%).

Model	Precision	Recall	F1	mIoU
DLinkNet	54.06	43.66	47.35	61.30
DeepLabV3+	64.64	40.46	45.87	62.44
NLLinkNet	78.52	41.22	53.04	64.65
RCFSNet	78.45	39.31	51.35	64.46
TCNet	58.94	50.15	53.14	65.22

**Table 3 sensors-25-01394-t003:** Quantitative results for the ablation study on DeepGlobe. The best results are highlighted in bold (%).

Baseline	CCA	CBAM	CCT	Precision	Recall	F1	mIoU
√				77.23	**80.29**	78.73	64.92
√	√			79.25	78.88	79.07	65.38
√	√	√		79.36	79.25	79.30	65.71
√	√	√	√	**79.51**	79.47	**79.57**	**66.08**

**Table 4 sensors-25-01394-t004:** Effect of different loss functions on road extraction results (%). The baseline model is U-Net. The best results are highlighted in bold(%).

	F1	mIoU
BCE loss	77.06	62.68
Dice loss	76.13	61.46
BCE loss + Dice loss	76.77	62.29
DBL	**78.09**	**64.06**

**Table 5 sensors-25-01394-t005:** Quantitative comparison on the DeepGlobe dataset (%). The model is TCNet.

	F1	mIoU
DeepGlobe	79.57	66.08
DeepGlobe + 1000 generated images	79.69	66.53

**Table 6 sensors-25-01394-t006:** Zero-shot on the Massachusetts Roads dataset (%). The model is TCNet.

Model	F1	mIoU
DeepGlobe	53.14	65.22
DeepGlobe +1000 generated images	54.57	66.03

## Data Availability

The data presented in this study are available upon request from the corresponding author.

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
