# Peer review of "Transferable Contextual Network for Rural Road Extraction from UAV-Based Remote Sensing Images"

_sensors, 2025, doi:10.3390/s25051394_

Round 1
Reviewer 1 Report
Comments and Suggestions for Authors
Abstract: acquiring UAV remote sensing data for rural areas is difficult, why? Do you mean obtaining the road information from UAV remote sensing data is difficult?
Introduction: why do you give the Fig. 1 in this section? Unclear. If you have done this work, why did you do? The location of Fig.1 is not good here.
The CBAM is proposed before, what is your innovation?
Why do you select the IoU as the main metric? MIoU or FwIoU is better.
In Figure 6, the road images displayed are all very simple images, which the human eye can even extract very quickly. I believe the authors should increase experiments, especially in areas with dense roads.
Author Response
Dear Reviewer,
Thank you for your thoughtful and constructive feedback. We appreciate the time and effort you have spent reviewing our manuscript. Your comments have helped us identify areas for clarification and improvement, and we have made the necessary revisions to address your concerns. Below, we provide detailed responses to each of the points you raised.
Comments 1:
Abstract: acquiring UAV remote sensing data for rural areas is difficult, why? Do you mean obtaining the road information from UAV remote sensing data is difficult?
Response 1:
We sincerely appreciate the reviewer's valuable feedback regarding the challenges of acquiring UAV remote sensing data for rural areas. In our original manuscript, we aimed to highlight the difficulties in obtaining high-quality UAV remote sensing data in rural regions, which is a critical prerequisite for accurate road extraction. However, we acknowledge that the phrasing in the abstract may have caused some ambiguity, and we have revised the text to clarify our intent.
In the revised abstract, we have explicitly stated that the challenges in rural road extraction stem not only from the complex characteristics of rural roads but also from the difficulty in acquiring UAV remote sensing data. Specifically, we have elaborated on the reasons why data acquisition is challenging in rural areas, including:
1.High cost of equipment: UAV and remote sensing equipment are expensive, limiting their widespread use in resource-constrained rural regions.
2.Extensive manual annotation: Rural roads often lack clear boundaries and are surrounded by complex terrain (e.g., vegetation, irregular shapes), making data annotation labor-intensive and time-consuming.
3.Limited regional policy support: Rural areas may lack the necessary policies or infrastructure to support UAV operations, such as flight permits or data-sharing agreements.
We hope that these revisions have addressed the reviewer's concerns and provided a clearer explanation of the challenges associated with acquiring UAV remote sensing data for rural areas. Thank you for pointing out this issue, as it has helped us improve the clarity and precision of our manuscript.
Comments 2:
Introduction: why do you give the Fig. 1 in this section? Unclear. If you have done this work, why did you do? The location of Fig.1 is not good here.
Response 2:
Thank you for your insightful comment. The purpose of Fig. 1 is to highlight the exceptional transferability of TCNet. This image, taken from the movie The Hobbit: An Unexpected Journey, is completely outside our training data. Yet, TCNet is able to segment roads effectively without any fine-tuning, demonstrating its robustness in unseen environments.
To clarify its relevance, we have revised the text in the Introduction to explicitly state why this figure is included at this stage. We believe this adjustment will make its placement more intuitive while maintaining the engaging and illustrative nature of the figure.
We appreciate your suggestion, which has helped us improve the clarity of our paper.
Comments 3:
The CBAM is proposed before, what is your innovation?
Response 3:
We thank the reviewer for raising this important question regarding the innovation of our approach. We acknowledge that the Convolutional Block Attention Module (CBAM) is an existing method proposed in prior work [42]. In our study, we did not modify the internal structure of CBAM but instead focused on its integration into our proposed framework to enhance the overall performance of rural road extraction.
The key innovation of our work lies in the novel combination of CBAM with other modules (e.g., the Clustered Contextual Transformer (CCT) and Clustered Cross Attention (CCA)) within the TCNet architecture. Specifically, we integrated CBAM into the skip connections between the encoder and decoder to improve feature representation and information transfer. This integration allows CBAM to enhance both spatial and channel attention, enabling the model to better capture road details in complex rural environments.
Comments 4:
Why do you select the IoU as the main metric? MIoU or FwIoU is better.
Response 4:
Thank you for your insightful comment. We apologize for the confusion caused by the omission of the "m" in the table. In fact, we have used mIoU (mean Intersection over Union) as the main evaluation metric in our experiments, which considers the IoU of all classes and provides a more comprehensive evaluation of the model’s performance across different classes. We have updated the tables to explicitly indicate that mIoU is used.
While FwIoU is indeed beneficial in situations where class imbalance is a concern, we opted for mIoU because it offers a more balanced performance evaluation across all classes. We hope this clarifies the choice of metric, and we have made the necessary revisions to reflect this more clearly in the paper.
Thank you for bringing this to our attention.
Comments 5:
In Figure 6, the road images displayed are all very simple images, which the human eye can even extract very quickly. I believe the authors should increase experiments, especially in areas with dense roads.
Response 5:
Thank you for your valuable feedback. We agree with your observation that the road images in Figure 6 are relatively simple, and testing the model on more complex and densely populated road areas would provide a better evaluation. In response to your suggestion, we have extended our experiments to include remote sensing images from the Alps mountain range, specifically in areas with dense road networks (images i and j in Figure 6). These images present more challenging scenarios, with intricate road patterns typical of mountainous regions, providing a better reflection of the model’s capability to handle complex and dense road structures.
In these more densely populated road areas, such as the Alps, we found that our model, TCNet, achieves performance comparable to DeepLabV3+ in terms of road extraction accuracy. However, TCNet requires approximately 16 million fewer parameters than DeepLabV3+, demonstrating the efficiency of our model in handling complex tasks with a lighter network architecture. We believe this further underscores the effectiveness of TCNet in maintaining high accuracy while significantly reducing model complexity.
Once again, we would like to thank you for your valuable suggestions, which have significantly improved the quality of the manuscript. We believe the revisions and clarifications we have provided address your concerns effectively, and we look forward to any further feedback you may have.
Best regards,
Renlong Wang

Reviewer 2 Report
Comments and Suggestions for Authors
The introduction effectively establishes the importance of the study; however, it would benefit from including additional references exploring recent advancements in attention mechanisms and modern data augmentation techniques, such as diffusion models and other generative approaches. The methodology is structured but lacks specific details on hyperparameters, including learning rates, batch sizes, and optimizer choices, which are essential for reproducibility. A more in-depth analysis of the individual contributions of the presented modules (CCT, CCA, and CBAM) would also strengthen the study. The results are well-presented, with precise figures and tables, but including more qualitative examples from diverse datasets could enhance the model's transferability. While the results support the conclusions, a more detailed discussion of the study's limitations and future research perspectives would be valuable. Finally, minor improvements in sentence structure and grammar, particularly in the methodology and discussion sections, would improve the clarity and readability of the manuscript.
Comments on the Quality of English LanguageThe quality of the English language in the manuscript is generally good and does not hinder the understanding of the research. However, minor improvements could enhance clarity and readability. Simplifying specific sentences would improve fluency, particularly in the methodology and discussion sections. These adjustments could further elevate the overall quality of the manuscript.
Author Response
Dear Reviewer,
Thank you for your constructive and thoughtful feedback. We greatly appreciate the time and effort you have spent reviewing our manuscript. Your suggestions have been extremely valuable in helping us enhance the clarity and depth of the study. In response to your comments, we have made several revisions to address the points you raised, particularly with respect to the methodology, hyperparameter details, and in-depth analysis of the individual contributions of the presented modules. We have also improved the readability of the manuscript by simplifying certain sentences, particularly in the Methodology and Discussion sections.
Reviewer’s Comment 1:
"The introduction effectively establishes the importance of the study; however, it would benefit from including additional references exploring recent advancements in attention mechanisms and modern data augmentation techniques, such as diffusion models and other generative approaches."
Response 1:
Thank you for your valuable suggestion. In response to your comment, we have expanded the Related Work section to include recent advancements in multi-scale segmentation designs and diffusion-based segmentation. Specifically, we added a discussion on multi-scale segmentation, including the Boundary-aware feature propagation method [34], which introduces a boundary-aware mechanism to enhance the segmentation of complex object boundaries in remote sensing images. This method improves segmentation performance by capturing features at multiple scales, which is particularly useful for tasks like road extraction. We also discussed the emerging role of diffusion models in segmentation tasks, highlighting the MaskDiffusion [27] approach, which utilizes pre-trained diffusion models to significantly improve segmentation accuracy, especially in complex backgrounds or scenarios with incomplete annotations. Additionally, we included Mosaicfusion [28], which employs diffusion models as data augmenters to improve large vocabulary instance segmentation, demonstrating their potential in enhancing segmentation performance across diverse datasets. These additions strengthen the background of our study by incorporating cutting-edge techniques in data augmentation and segmentation, aligning our work with the latest trends in the field. We appreciate your feedback, which allowed us to broaden the scope of the related work and better contextualize our approach within current advancements.
Reviewer’s Comment 2:
The methodology is structured but lacks specific details on hyperparameters, including learning rates, batch sizes, and optimizer choices, which are essential for reproducibility.
Response 2:
Thank you for your insightful comment regarding the missing details on hyperparameters. In response, we have added a more comprehensive description of the key hyperparameters used during the training of the TCNet model. Specifically, the batch size was set to 2 images per GPU, and the learning rate was initially set to 0.0002 for the first 150 epochs, which was then decayed to 0.00002 for the remaining epochs. We employed the Adam optimizer with parameters β₁ = 0.9 and β₂ = 0.999. Furthermore, early stopping based on validation loss was applied to prevent overfitting, ensuring that the best-performing model was saved. Additionally, to provide a reference for hardware performance, we included a Gfops analysis, where a benchmark run on the NVIDIA GeForce RTX 3060 GPU resulted in an estimated value of 0.085756 Gfops. We believe these additions will improve the clarity and reproducibility of the methodology. Thank you again for your valuable suggestion.
Reviewer’s Comment 3:
A more in-depth analysis of the individual contributions of the presented modules (CCT, CCA, and CBAM) would also strengthen the study.
Response 3:
Thank you for your valuable feedback! We have added a detailed improvement analysis of the Clustered Contextual Transformer (CCT) block in the method section. In the newly added paragraph, we thoroughly explain how the CCT overcomes the limitations of traditional convolutional self-attention mechanisms in rural road extraction by leveraging localized group convolutions and a context-aware attention mechanism. Specifically, we describe how CCT divides the feature map into clusters, enabling the model to focus on region-specific features while simultaneously capturing global contextual information. This approach significantly improves the model’s ability to interpret complex road patterns, especially in rural areas where road features are non-uniform and interwoven.
The key contribution of CCT lies in its clustering strategy, which allows the model to focus on learning specific features for different regions, thereby effectively handling the complex and irregular patterns typical in rural terrains. We believe this addition now provides a clearer explanation of how the CCT block contributes to enhancing model performance, as well as its improved transferability and robustness in rural road segmentation tasks.
Reviewer’s Comment 4:
The results are well-presented, with precise figures and tables, but including more qualitative examples from diverse datasets could enhance the model's transferability.
Response 4:
Thank you for your valuable feedback and suggestion. We understand your point about including more qualitative examples from diverse datasets to enhance the model's transferability. At present, however, the main remote sensing road segmentation datasets for rural areas are primarily limited to the Massachusetts dataset, the DeepGlobe dataset, and the Rural dataset.
In response to your suggestion, we have expanded our experiments to include images from the Alps region, specifically focusing on areas with dense road networks, which are shown in images i and j of Figure 6. These images offer a more complex and challenging setting and help demonstrate the model's transferability to regions with more densely populated road networks.
We hope this addition addresses your concern and further illustrates the model’s robustness. Looking ahead, we plan to create our own remote sensing road dataset focused on highland regions, which will help overcome potential data bottlenecks in road extraction tasks and allow for more diverse testing and validation. This will enable us to further evaluate the model’s generalization capability across different terrains.
Reviewer’s Comment 5:
While the results support the conclusions, a more detailed discussion of the study's limitations and future research perspectives would be valuable.
Response 5:
Thank you for your valuable feedback and suggestions. In response to your comment about providing a more detailed discussion of the study's limitations and future research perspectives, we have made the following revisions:
In the Discussion section, we have expanded our analysis to more thoroughly address the limitations of our study. We now discuss challenges such as the model's performance in complex urban environments, handling highly occluded roads, and dealing with deep shadows. We also highlight issues related to artifacts in the images generated by the stable-diffusion-2-inpainting model, as shown in Figure 9, and acknowledge the challenges of ensuring accurate labels in datasets like DeepGlobe. Additionally, we emphasize the need for more diverse datasets, especially for highland regions, to overcome data scarcity and improve generalization.
In the Conclusion section, we have extended our discussion on future research directions. Specifically, we explore the potential for improving the model’s robustness in urban environments, integrating multi-scale fusion techniques, and building a highland-region road dataset to alleviate data bottlenecks in road extraction tasks. These future efforts will help address existing limitations and further enhance the model's applicability across diverse environments.
Reviewer’s Comment 6:
Finally, minor improvements in sentence structure and grammar, particularly in the methodology and discussion sections, would improve the clarity and readability of the manuscript.
Response 6:
Thank you for your helpful suggestion. In response, we have made several improvements to the sentence structure and grammar, particularly in the Methodology and Discussion sections, to enhance the clarity and readability of the manuscript. These revisions aim to streamline the explanations, reduce redundancy, and ensure a more direct and clear presentation of our ideas. For example, we refined the wording of certain sentences, such as simplifying complex expressions and rephrasing passive constructions into more active forms, making the content easier to follow. Additionally, we corrected minor grammatical errors and improved the flow of ideas to strengthen the overall readability. We believe these changes have improved the presentation of the paper, making it clearer and more accessible for the reader.
Once again, we thank you for your insightful suggestions, which have significantly contributed to the improvement of the manuscript. We believe that the revisions we have made, including the addition of references to recent advancements in attention mechanisms and data augmentation techniques, as well as the clarification of hyperparameters and a more detailed discussion of the study's limitations, have strengthened the manuscript. We hope that the changes adequately address your concerns and that the revised version meets your expectations. We look forward to any further feedback you may have.
Best regards,
Renlong Wang

Reviewer 3 Report
Comments and Suggestions for Authors
Strength:
1, Overall writing is good and easy to follow.
2, The proposed TCNet achieves SOTA performance on several road segmentation datasets.
3, The proposed clustered contextual transformer block is interesting and effective according to the experiment.
Weakness:
1, The details of Stable-diffusion-2-inpainting for Data Augmentation are not that clear since this is an important contribution to this work.
2, The improvement analysis of clustered contextual transformer block is missing.
3, Several related works are missing, including diffusion-based segmentation (mask generation) and multi-scale segmentation designs.
[-] MaskDiffusion: Exploiting Pre-trained Diffusion Models for Semantic Segmentation, arxiv-2024.
[-] Mosaicfusion: Diffusion models as data augmenters for large vocabulary instance segmentation, IJCV-2024.
[-] Boundary-aware feature propagation for scene segmentation, ICCV-2019
4, Missing Gfops and parameter analysis.
Author Response
Dear Reviewer,
Thank you for your constructive feedback and thoughtful suggestions. We appreciate your positive comments on the overall quality of the writing, the proposed TCNet model, and the clustered contextual transformer block. We also appreciate your thorough review, which has highlighted areas for improvement. We have carefully addressed each of your points and have made the necessary revisions to improve the clarity and depth of the paper.
Comments 1:
The details of Stable-diffusion-2-inpainting for Data Augmentation are not that clear since this is an important contribution to this work.
Response 1:
We appreciate the reviewer’s comment and have expanded the explanation of the Stable Diffusion inpainting technique used for data augmentation. We clarified how the model selectively modifies only the background elements of the input images while preserving the road structures, ensuring that the augmented data is both realistic and diverse. Additionally, we provided a more detailed description of the methodology, including the use of the OpenCLIP-ViT encoder to guide the diffusion process and how this approach addresses the data bottleneck problem in remote sensing datasets. The enhanced explanation should make the technique’s contribution to the overall task clearer. We hope this revision improves the clarity and transparency of our approach.
Comments 2:
The improvement analysis of clustered contextual transformer block is missing.
Response 2:
Thank you for your valuable feedback! We have added a detailed improvement analysis of the Clustered Contextual Transformer (CCT) block in the method section. In the newly added paragraph, we thoroughly explain how the CCT overcomes the limitations of traditional convolutional self-attention mechanisms in rural road extraction by leveraging localized group convolutions and a context-aware attention mechanism. Specifically, we describe how CCT divides the feature map into clusters, enabling the model to focus on region-specific features while simultaneously capturing global contextual information. This approach significantly improves the model’s ability to interpret complex road patterns, especially in rural areas where road features are non-uniform and interwoven.
The key contribution of CCT lies in its clustering strategy, which allows the model to focus on learning specific features for different regions, thereby effectively handling the complex and irregular patterns typical in rural terrains. We believe this addition now provides a clearer explanation of how the CCT block contributes to enhancing model performance, as well as its improved transferability and robustness in rural road segmentation tasks.
Comments 3:
Several related works are missing, including diffusion-based segmentation (mask generation) and multi-scale segmentation designs.
[-] MaskDiffusion: Exploiting Pre-trained Diffusion Models for Semantic Segmentation, arxiv-2024.
[-] Mosaicfusion: Diffusion models as data augmenters for large vocabulary instance segmentation, IJCV-2024.
[-] Boundary-aware feature propagation for scene segmentation, ICCV-2019
Response 3:
Thank you for your insightful comment. We appreciate the suggestion to include more related works, particularly those involving diffusion-based segmentation and multi-scale segmentation designs. We have carefully revised the Related Work section to address this, and the following works have now been incorporated:
MaskDiffusion [27], which exploits pre-trained diffusion models for semantic segmentation, is now discussed to highlight its effectiveness in improving segmentation accuracy, particularly in complex backgrounds or scenarios with incomplete annotations. We emphasize how MaskDiffusion enhances traditional models by generating high-quality segmentation masks through a reverse diffusion process.
Mosaicfusion [28], which uses diffusion models as a data augmentation tool for large-scale instance segmentation, is also mentioned. This work enhances the performance of instance segmentation models in large-vocabulary tasks by generating diverse and realistic data samples, which is especially relevant in rural road extraction where dataset variability can be a challenge.
The Boundary-aware feature propagation method [34], introduced for scene segmentation, is now included to highlight the importance of multi-scale strategies in improving feature propagation in complex boundaries, particularly in road extraction tasks where the road boundaries may be intricate and irregular.
We hope this revision adequately addresses your comment and enriches the Related Work section by including these important references.
Comments 4:
Missing Gfops and parameter analysis.
Response 4:
Thank you for your valuable feedback. In response to your comment regarding the Gfops and parameter analysis, we have made the following revisions:
Gfops Calculation: We have added a Gfops estimation to provide a reference for the computational capability of the hardware used in our experiments. However, we would like to clarify that this Gfops value was obtained through a simple benchmark test on the NVIDIA GeForce RTX 3060 GPU and is not directly linked to the exact computational requirements of the model during training. The estimated Gfops value is 0.085756 Gfops based on a benchmark involving 10,000,000 floating point operations, which provides a rough idea of the GPU's performance for computational tasks.
Parameter Analysis: We have also provided a detailed description of the training parameters, including the batch size, optimizer settings, learning rate schedules, data augmentation techniques, and hardware configuration. Additionally, we explained the implementation of early stopping to prevent overfitting and ensure that the best-performing model was selected based on validation loss.
We hope these revisions address your concerns and contribute to a clearer understanding of the computational performance and training setup. Thank you again for your insightful suggestions.
Once again, we greatly appreciate your insightful feedback, which has undoubtedly contributed to the enhancement of the manuscript. We hope the revisions and clarifications we have provided address your concerns effectively. We look forward to your further feedback.
Best regards,
Renlong Wang

Round 2
Reviewer 3 Report
Comments and Suggestions for Authors
The authors have solved mu questions.